# Anti-U11/U12 Antibodies as a Rare but Important Biomarker in Patients with Systemic Sclerosis: A Narrative Review

**DOI:** 10.3390/diagnostics13071257

**Published:** 2023-03-27

**Authors:** Marvin J. Fritzler, Chelsea Bentow, Lorenzo Beretta, Boaz Palterer, Janire Perurena-Prieto, Maria Teresa Sanz-Martínez, Alfredo Guillen-Del-Castillo, Ana Marín, Vicent Fonollosa-Pla, Eduardo Callejas-Moraga, Carmen Pilar Simeón-Aznar, Michael Mahler

**Affiliations:** 1Cumming School of Medicine, University of Calgary, Calgary, AB T2N 4N1, Canada; 2Research and Development, Werfen, Autoimmunity Headquarters and Technology Center, San Diego, CA 92131-1638, USA; 3Scleroderma Unit and (Referral) Center for Systemic Autoimmune Diseases, Fondazione IRCCS Ca’ Granda Ospedale Maggiore Policlinico di Milan, 20122 Milano, Italy; 4Department of Experimental and Clinical Medicine, University of Florence, 50121 Firenze, Italy; 5Department of Immunology, Hospital Universitari Vall d’Hebron, Universitat Autònoma de Barcelona, 08193 Barcelona, Spain; 6Unit of Systemic Autoimmune Diseases, Department of Internal Medicine, Hospital Universitari Vall d’Hebron, 08035 Barcelona, Spain; 7Department of Internal Medicine, Hospital Público de Monforte, 27400 Lugo, Spain

**Keywords:** systemic sclerosis, U11/U12, autoantibodies, interstitial lung disease, SSc

## Abstract

Anti-nuclear (ANA) are present in approximately 90% of systemic sclerosis (SSc) patients and are key biomarkers in supporting the diagnosis and determining the prognosis of this disease. In addition to the classification criteria autoantibodies for SSc [i.e., anti-centromere, anti-topoisomerase I (Scl-70), anti-RNA polymerase III], other autoantibodies have been associated with important SSc phenotypes. Among them, anti-U11/U12 ribonucleoprotein (RNP) antibodies, also known as anti-RNPC-3, were first reported in a patient with SSc, but very little is known about their association and clinical utility. The U11/U12 RNP macromolecular complex consists of several proteins involved in alternative mRNA splicing. More recent studies demonstrated associations of anti-anti-U11/U12 antibodies with SSc and severe pulmonary fibrosis as well as with moderate to severe gastrointestinal dysmotility. Lastly, anti-U11/U12 autoantibodies have been strongly associated with malignancy in SSc patients. Here, we aimed to summarize the knowledge of anti-U11/U12/RNPC-3 antibodies in SSc, including their seroclinical associations in a narrative literature review.

## 1. Introduction

Systemic sclerosis (SSc, also known as scleroderma) is a chronic autoimmune disease that affects connective tissue and can cause a wide range of symptoms. The main causes of mortality in SSc are cardiovascular disease, interstitial lung disease (ILD), and renal crisis [1]. Additionally, other complications such as pulmonary hypertension, gastrointestinal dysfunction, and stroke can also contribute to mortality in individuals suffering from SSc. Anti-nuclear (ANA) or anti-cellular antibodies (ACA) are identified in approximately 90% of SSc patients and are key biomarkers in supporting the diagnosis and determining the prognosis of SSc [1,2,3]. As well as the more well-recognized autoantibodies that are included in the 2013 American College of Rheumatology (ACR)/European League Against Rheumatism (EULAR) classification criteria for SSc [3] (i.e., anti-centromere, anti-topoisomerase I (Scl-70 or ATA), and anti-RNA polymerase III), other autoantibodies have been associated with important SSc phenotypes (Table 1) [1]. In addition to these well-known autoantibodies, several other antibodies have been specifically associated with SSc, including anti-U3 ribonucleoprotein (RNP) or, more specifically, anti-fibrillarin [4,5,6], anti-Th/To [7,8,9,10,11], anti-eukaryotic initiation factor 2B (eIF2B) [12], anti-RuvBL1/2 [12,13], and anti-TERF-1 antibodies [14,15]. In addition to those SSc-specific antibodies (SSc-SA), a wide range of SSc-associated antibodies have also been reported over the last decades: Anti-U1 RNP, anti-PM–Scl [16,17,18,19], anti-Ku [20], anti- Ro52/tripartite motif (TRIM) 21, and anti-human upstream binding factor (hUBF)/anti-NOR-90 antibodies [21]. In myositis, the definition of autoantibody specificities is more established. Antibodies that mostly occur in myositis are referred to as myositis specific antibodies or MSA, while antibodies that occur in myositis, but to a certain extent also in other conditions, are termed myositis associated antibodies (MAA). Here, we aim to introduce a similar nomenclature for SSc, namely SSc-specific (SSc-SA) and SSc-associated antibodies (SSc-AA). While it is well established that the classification criteria markers belong to the group of SSc-AA, for some antibodies, more studies are needed to conclude if they belong to the SSc-SA or SSc-AA group.

Anti-U11/U12 RNP (also referred to as RNPC-3) antibodies were first reported in a SSc patient [22] in 1993, but very little is known about their association and utility [23]. The U11/U12 RNP macromolecular complex consists of several proteins and is involved in alternative mRNA splicing. Here, we aim to summarize the knowledge of anti-U11/U12/RNPC-3 antibodies in SSc, including their serological and clinical associations in a narrative literature review.

**Table 1 diagnostics-13-01257-t001:** Overview of autoantibodies in systemic sclerosis.

	Antigen	Prevalence	IIF	Comments (Main Antigens)
SSc-SA Included in ACR/EULAR classification criteria	Topoisomerase I (Scl-70)	20–30%	AC-29	DNA topoisomerase I
	Centromere [24]	25–50%	AC-03	CENP-A, CENP-B
	RNA Polymerase III (RNAP3)	10–20%	AC-10/ AC-04/05	Nucleolar pattern is not reliable in detecting RNAP3, often a speckled pattern prevails
SSc-SA	Fibrillarin [4,6]	3–20%	Nu (AC-09)	U3 RNP
	Th/To [7,8,9,10,11]	3–10%	Nu (AC-08)	Rpp25, Rpp38
	RuvBL1/2 [8,9,12,13]	1–3%	Sp	Double hexamer of RuvBL1 and RuvBL2 proteins
SSc-AA	U1 RNP	5–15%	Sp (AC-05)	U1-RNP 70 kDa, U1 RNP-A and U1 RNP-C
	Ku [20]	3–10%	Sp (AC-04)	80- and 70-kDa binding dimeric protein
	PM/Scl [18]	5–10%	Nu (AC-08)	PM/Scl-100 protein of human exosome
	Ro52 [25,26,27,28]	15–25%	?	Ro52/tripartite motif (TRIM) 21
TBD	U11/U12 RNP [22,29]	3–5%	Sp (AC-02/04/05)	U11/U12 RNP complex of spliceosome. RNPC-3 is the 65 kDa main target
	TERF-1 [14,15]	3–5%	TBD	Shelterin protein
	eIF2B [12]	1–3%	Cyto	Eukaryotic initiation factor 2B
	NOR 90 [21,30]	<5%	AC-10	human upstream binding factor (hUBF)
	B23 [30,31,32]	<5%	AC-08	Nucleophosphmin

AC = anti-cell pattern by ICAP designation; NOR = nucleolar organizer region; Nu = nucleolar staining pattern; RNAP = RNA polymerase; RNP = ribonucleoprotein; Sp = speckled nuclear staining pattern; SSc-SA = systemic sclerosis specific antibodies; SSc-AA = systemic sclerosis associated antibodies; TBD = to be determined; TERF-1 = telomeric repeat-binding factor 1.

## 2. Methods

A search of Medline and Embase up to September 2022 was performed using the medical subject heading terms “anti-U11/U12”, “anti-RNPC-3”, “systemic sclerosis”, “scleroderma”, “connective tissue disease”, to identify publications. Manual searches of references cited in the retrieved articles were also performed. The eligibility criteria included: (1) studies assessing anti-U11/U12 RNP antibodies in patients with SSc; (2) studies assessing anti-U11/U12 RNP antibodies in patients with connective tissue diseases (CTDs); and (3) only peer-reviewed publications written in English and involving human subjects. Abstracts were excluded. No restriction on time was applied. Due to the limited number of retrieved publications, a narrative literature review was conducted.

## 3. Historical Perspective

Although first described almost 30 years ago by Gilliam and Steitz [22], very little is known about the HEp-2 indirect immunofluorescence (IIF) pattern associated with anti-U11/U12 RNP autoantibodies (Figure 1). In the seminal paper, the index patient (Ru) was a 40-year-old Caucasian female who presented with Raynaud phenomenon followed by rapidly progressive cutaneous SSc involving her face, trunk, and extremities accompanied by mild restrictive lung disease and esophageal dysmotility (Table 2). The HEp-2 IIF result was reported as a titer of 1:650 with a nuclear speckled pattern “resembling” the IIF pattern “associated with anti-Sm antibodies”. ANA analysis showed an “anti-extractable nuclear antigen” in the absence of antibodies directed to double-stranded DNA, Scl-70, SS-A/Ro, or SS-B/La by ELISA. Using immunoprecipitation (IP) assays, the patient’s serum was found to have antibodies directed to U11/U12 RNP.

More recently, the screening of large cohorts of patients revealed that anti-U11/U12 RNP antibodies were present at low frequency (3.2–8.0%), but with high disease specificity for SSc. Among the components of the U11/U12 RNP complex, the 65 kDa RNA binding domain containing 3 (RNPC-3) protein was defined as an immunodominant target of human autoantibodies [22,29,33,34]. RNPC-3 is part of the minor spliceosome complex responsible for removing U12-type introns from pre-messenger RNA and is comprised of two RNA recognition motifs that likely contact one of the small nuclear RNAs [35].

**Figure 1 diagnostics-13-01257-f001:**
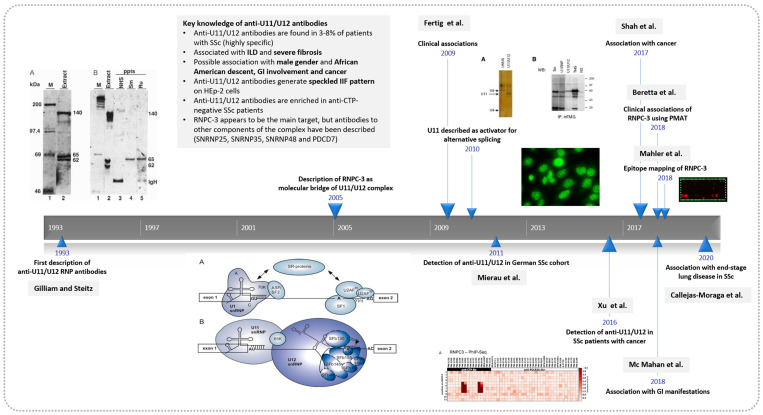
History of anti-U11/U12 RNP autoantibodies. The history of anti-U11/U12 RNP antibodies started in 1993 with the discovery of the antibodies by Gilliam and Steitz [22]. During the following years, several studies focused on the molecular aspects and cellular function of the U11/U12 RNP complex. It was not until 2009 when Fertig et al. [36] added to the knowledge of clinical associations of anti-U11/U12 RNP. Later on, Shah et al. first reported on a potential association with cancer and McMahan et al. [37] with gastrointestinal involvement. Other key studies on anti-U11/U12 RNP antibodies are included [29,33,34,38,39].

## 4. Detection Methods

In accord with the limited number of studies published on anti-U11/U12 antibodies, a narrow range of methods are available for their detection. However, in the US, some specialized laboratories may offer anti-U11/U12 antibody testing under the umbrella of a laboratory developed test (LDT) [40].

### 4.1. Immunoprecipitation

As referenced above, anti-U11/U12 antibodies can be detected by IP. Early studies used cell extracts for IP while most recent studies relied on IP of RNPC-3 using *in-vitro* translated products of the cognate cDNA. Using cellular extracts can be challenging since many proteins have a molecular mass approximating that of RNPC-3 producing many bands of approximately 65 kDa. Therefore, it has been important to develop a solid phase immunoassay using the specific antigenic target of anti-U11/U1 RNP antibodies.

### 4.2. Indirect Immunofluorescence

According to Gilliam and Steitz, anti-U11/U12 RNP antibodies were associated with an IIF pattern that resembled the coarse speckled nuclear pattern (RNP like) without staining of the chromatin region [22]. In order to better characterize the characteristics of anti-U11/U12 RNP antibodies, in a previous study we identified monospecific anti-U11/U12 positive serum samples in a SSc cohort that used a particle-based multi-analyte technology (PMAT) for the detection of anti-U11/U12 antibodies. Two serum samples with clear monospecific reactivity to U11/U12 were selected and subsequently used for IIF testing (Figure 2). We assessed the commutability of four different HEp-2 IIF substrates (BioRad, Inova Diagnostics Inc., ImmunoConcepts, Binding Site) in detecting anti-U11/U12 RNP (RNPC-3) antibodies in human sera. The IIF pattern was a fine discrete nuclear speckled staining that had features similar to ICAP AC-02 (dense fine speckles), AC-04 (fine speckles), or AC-05 (large coarse speckles). However, the anti-U11/12 exhibited distinctive IIF staining features such as somewhat larger and more rounded, discrete nuclear speckles (compared to AC-02 and AC-04) that were smaller and more clearly defined than those typically associated with anti-U1 RNP and anti-Sm (i.e., AC-05), a finding in keeping with previous reports [22,39,41]. Depending on the commercial source of the HEp-2 substrate, variable staining of the nucleolar region, intense staining of the chromatin and/or perichromatin region of metaphase cells, and faint cytoplasmic speckled staining was also observed. When serial serum dilutions were used to analyze the end point titers, all four kits used were within one end-point dilution (i.e., 1:640–1:1280).

In a preliminary study, co-localization by IIF of human anti-U11/U12 (RNPC-3) with rabbit anti-RNPC-3 (rabbit polyclonal) showed significant overlap between the structures targeted by the two antibody populations [39]. Further experiments are warranted to understand the co-localization with other autoantigens including but not limed to [PML, coilin/p80 and Survival of Motor Neuron (SMN) Complex].

ICAP [1] and others [19] recommend reflex testing of samples with AC-04 or AC-05 IIF patterns to assays containing known autoantigens, including RNP/Sm, Ro60/SSA, and La/SS-B. However, based on our observation, inclusion of anti-U11/U12 (RNPC-3) in reflex testing algorithms should be considered when HEp-2 IIF patterns resembling the AC-02, AC-04, or AC-05 are observed and clinical suspicion of SSc is present, especially when no antibodies to RNP/Sm, Ro60/SSA, and La/SS-B can be detected. In addition, since the HEp-2 IIF pattern is not identical to AC-04 or AC-05, the pattern associated with anti-RNPC-3 antibodies might be considered in future efforts of ICAP to define new clinically relevant patterns (expert level). Whether this is applicable to clinical practice is unlikely, considering the possibility of mixed patterns and the rarity of these antibodies. This is important as the HEp-2 IIF is still used as the first line screening test for patients suspected of having SSc [13]. Reflexing strategies taking into consideration both the clinical suspicion and the IIF pattern might be necessary in order to appropriately implement testing for anti-U11/U12 (RNPC-3) antibodies in the clinical routine. Landon-Cardinal et al. in a cohort of patients with scleromyositis, recognized three novel clinico-serological subsets, including a group of patients with a speckled IIF pattern [42].

### 4.3. Particle-Based Multi-Analyte Technology (PMAT)

Recently, a novel PMAT system has been developed that allows for the detection of antibodies to RNPC-3 as a molecular surrogate for the U11/U12 RNP complex [29,33,34]. The PMAT assay utilizes paramagnetic particles with unique signatures and a digital interpretation system. More specifically, antigens are coupled to paramagnetic particles that carry unique signatures and are incubated with diluted patient serum samples. After 9.5 min incubation at 37 °C, particles are washed and incubated 9.5 min incubation at 37 °C with anti-human IgG conjugated to phycoerythrin (PE) to label the bound autoantibodies. After a final wash cycle, fluorescent signal intensity on the particles is captured using a digital imager and analyzed using proprietary algorithms to extract meaningful information for each analyte.

## 5. Co-Existence of Other Antibodies

Due to the limited number of studies available and the rarity of anti-U11/U12 antibodies, the association of anti-U11/U12 antibodies with other SSc-SA or SSc-AA is not well studied. Like other SSc-SA, anti-U11/U12 antibodies tend to be exclusive (occurring without other SSc antibodies). In a recent study, the prevalence and levels of anti-RNPC-3 antibodies were higher in anti-Scl-70, ACA, and/or anti-RNA Pol III triple negative patients [10/106 (9.4%) vs. 6/193 (3.1%), *p* = 0.03; and median titers 122.0 vs. 107.5, *p* = 0.08] [39]. However, further studies might show coexistence of anti-U11/U12 with ACA, anti-Scl-70, anti-RNA Pol III, or other SSc antibodies.

## 6. Clinical Associations

Anti-U11/U12 (RNPC-3) antibodies have been associated with ILD and pulmonary fibrosis. When compared to other SSc-SA, anti-U11/U12 showed significantly stronger association with pulmonary fibrosis versus all other antibodies studied (anti-centromere, anti-PM/Scl, anti-RNA Pol III, anti-Th/To, anti-Scl-70, anti-U1 RNP, and anti-U3 RNP) with the exception of anti-Ku for which it is likely the number of cases was too low to reach statistical significance [36]. In addition, associations were reported with moderate to severe gastrointestinal (GI) dysmotility and severe Raynaud’s phenomenon [37]. The association with GI involvement is interesting as RNPC-3 deficient mice demonstrate arrested development and impaired gastrointestinal homeostasis [43]. In addition to those clinical associations, in a study of 318 SSc patients with cancer, Shah et al. reported that anti-RNPC-3 antibodies were found in 3.8% of the patients and were associated with a short cancer-scleroderma interval (median 0.9 years). In addition, 33% had an associated myopathy [44].

To further analyze the clinical associations of anti-RNPC-3 antibodies in SSc, sera from 88 SSc patients were studied of which 32 cases had documented cancer. Sera from 64 systemic lupus erythematosus (SLE) patients, 65 idiopathic inflammatory myopathies (IIM), and 20 health individuals (HI) were included as a comparator group [39]. The frequency of autoantibodies to both recombinant RNPC-3 and a synthetic peptide derived from the RNPC-3 sequence tended to be higher in SSc and SLE compared to IIM and HI [39]. When the SSc cancer patients were compared with an unselected SSc cohort, no significant difference was found [39]. This finding appears to be inconsistent with previous findings [44], but might be explained by the different selection process of the samples. Possible explanations likely include the cancer type and the time difference between cancer onset and blood sampling. Further validation studies of large SSc cohorts are needed to determine if anti-U11/U12 RNP antibodies are associated with cancer and/or GI involvement.

The most recent study on anti-U11/U12 (RNPC-3) antibodies included a cohort of 447 SSc patients from Barcelona (n = 286) and Milan (n = 161) [29]. All samples were tested using the PMAT assay using recombinant RNPC-3 as the antigen. Anti-U11/U12 (RNPC-3) positive and negative patients were compared in terms of clinical presentations. Epidemiological, clinical features, and survival were analyzed. End-stage lung disease (ESLD) was defined if the patient had a forced vital capacity < 50% of predicted and needed oxygen therapy or lung transplantation. Event-free survival (EFS) was defined as the period of time free of either ESLD or death. A total of 19 of 447 (4.3%) patients had anti-U11/U12 (RNPC-3) antibodies and ILD was more frequent (11, 57.9% vs. 144, 33.6%, *p* = 0.030) in individuals with anti-RNPC-3 antibodies. More patients reached ESLD in the positive group (7, 36.8% vs. 74, 17.3%, *p* = 0.006), and a higher use of non-glucocorticsteroid immunosuppressive drugs was observed (11, 57.9% vs. 130, 30.4%, *p* = 0.012). Anti-U11/U12 (RNPC-3) positive patients had lower EFS, both in the total cohort (log-rank *p* = 0.001) as well as in patients with ILD (log-rank *p* = 0.002). In multivariate Cox regression analysis, diffuse cutaneous subtype, age at onset, the presence of ILD or pulmonary arterial hypertension, and the expression of anti-RNPC-3 positivity or anti-Scl-70 antibodies were independently associated with worse EFS. The presence of anti-U11/U12 (RNPC-3) was associated with higher frequency of ILD and either ESLD or death. These data suggest that anti-RNPC-3 are independently associated with a poor prognosis in SSc, especially if ILD is a co-morbid feature. In summary, evidence is mounting that anti-U11/U12 (RNPC-3) antibodies represent a rare but important antibody specificity as they often occur in the absence of other SSc-related antibodies and are detected in patients with ILD [36].

Recently, using a bead-based antigen array, high levels of anti-U11/U12 antibodies were described in patients hospitalized with COVID-19 [45]. Although the findings are worth mentioning in the context of this review, further studies are required to interpret the data. No validation data for anti-U11/U12 antibodies on the method used for the study by Chang et al. [45] was available.

**Table 2 diagnostics-13-01257-t002:** Overview of studies on anti-U11/U12 antibodies.

References	No. Positives	Method/Antigen	Comments
Gilliam & Steitz 1993 [22]	1 patient	IP with HeLa extract	Initial description of anti-U11/U12 antibodies. Patient Ru
Fertig et al. 2009 [36]	33 U11/U12 positive 15/462 (3.2%) in SSc	IP, RT-PCR and SB	Association with GI involvement (*p* = 0.01), pulmonary fibrosis (*p* = 0.0001) and mortality (*p* = 0.01)
Mierau et al. 2011 [46]	1/863 (0.1%) of SSc patients defined as anti-U11/U12 positive	IP + IIF	An RNP-like IP pattern and coarsely speckled ANA IIF, without any U1-RNP signals in line assay and ID; U11-RNP specificity detected by C Will and R Lührmann, Marburg, Germany
Xu et al. 2016 [38]	4/16 (25%) CTP-negative SSc patients with cancer positive for anti-U11/U12	IP PhIP-Seq PLATO-BC	Detection of antibodies to other components of the U11/U12 complex
Shah et al. 2017 [44]	12/318 (3.8%) in SSc	IP with RNPC-3	NOTE: all of the SSc had cancer
Beretta et al. 2018 [33]	19/613 (3.1%) SSc patients positive for anti-U11/U12	PMAT	Prevalence of anti-RNPC-3 antibodies in an Italian and Spanish cohort of SSc patients Strong association with ILD 13/19 CTP-negative
Mahler et al. 2018 [34]	3/90 (3.3%) SSc patients positive for anti-U11/U12	PMAT/SPPA	Epitope mapping identified several linear epitopes on RNPC-3
Mahler et al. 2020 [39]	10/106 (9.4%) CTP-negative SSc patients positive for anti-U11/U12	PMAT	No association with cancer observed
McMahan et al. 2018 [37]	6 anti-RNP-C positive	IP with RNPC-3	Association with male gender and African American decent Association with moderate-to-severe GI disease and ILD
Callejas-Moraga et al. 2021 [29]	19/447 (4.3%) in SSc	PMAT RNPC-3	Association with severe ILD
Chang et al. [45]	COVID-19 patients (no cut-off provided)	Multiplex	Individual COVID-19 patients with high levels of anit-U11/U12 antibodies

ILD = interstitial lung disease; IP = immunoprecipitation; GI = gastrointestinal; PMAT = particle-based multi-analyte technology; SB = southern blotting; RT-PCR = reverse transcription-polymerase chain reaction; SPPA = Solid phase peptide arrays.

## 7. Biochemical Aspects and Immunogenicity

The U11/U12 complex consists of at least eight U11/U12 RNP specific proteins and is involved in alternative splicing [35,47]. Members of the complex include hPrp43 (DHX15), RNPC-3, PDCD7, snRNP48, snRNP35, ZCRB1, snRNP25, and ZMAT5 (see Table 3). In addition, Sm proteins and the SF3b complex join the U11/U12 specific proteins in the complex. The U11/U12 snRNP 65 kDa protein (or RNPC-3) acts as a molecular bridge, binding the U12 snRNA and U11-59 kDa protein [48]. Although RNPC-3 has been defined as a major antigenic target of the complex, systematic analyses have not been performed, especially with newer approaches and technologies. However, in a recent study, antibodies to other components of the U11/U12 complex were identified [38]. Bacteriophage Immunoprecipitation Sequencing (PhIP-Seq) identified SNRNP48 and PDCD7 as antigenic targets and barcoded parallel analysis of translated open reading frames (PLATO-BC) detected immune reactivity to SNRNP25, SNRNP35 in addition to RNPC-3.

## 8. Epitope Mapping of RNPC-3

The epitope distribution of anti-U11/U12 antibodies was analyzed in two studies [34,38]. In 2016, Xu et al. used phage-immunoprecipitation sequencing to study the antibody response in SSc patients with cancer and observed both intra-and inter-molecular epitope spreading [38]. In 2018, the second study utilizing solid phase peptides revealed several linear epitopes across the entire protein [34]. The identified candidate epitopes were subsequently utilized to synthesize synthetic, biotinylated, and soluble peptides for testing using the novel PMAT described above. The reactivity to recombinant RNPC-3 (rRNPC-3) and to the RNPC-3 derived peptide (pRNPC-3) was correlated in two independent cohorts. In the first study using samples from SSc, IIM, and SLE patients and healthy individuals, a high level of correlation was observed (Spearman’s rho = 0.64, 95% Confidence interval 0.58–0.69; *p* < 0.0001) [34]. In the second study of 299 patients, the correlation was less pronounced but was still highly significant (Spearman = 0.32, *p =* 0.0001; chi-squared *p =* 0.0002) [39]. Immunoadsorption showed significant inhibition using the RNPC-3 derived peptide, but not with a control peptide [39]. Taken together, these data indicate that the identified epitope represents a major determinant of the B-cell immune response towards RNPC-3. Future studies are warranted to evaluate if the identified peptides representing the key epitopes might be used as molecular surrogates for the detection of anti-U11/U12 antibodies or if they identify more specific subsets of patients.

## 9. Implication for New Treatment Strategies

In contrast to traditional treatment strategies, modern approaches are targeting specific clinical presentations and organ involvement [1]. During the last years, special focus was placed on lung disease in SSc patients. Consequently, biomarkers such as anti-U11/U12 antibodies might be useful to identify patients with rapidly progressing ILD [1,49], especially since the classification criteria marker anti-Scl-70, ACA and anti-RNA Pol III do not effectively allow for stratification, albeit still being useful f identify patients at risk for SSc-ILD [49,50,51]. Similarly, it must be determined to what extent these antibodies, in conjunction with other risk factors for SSc-ILD, such as the diffuse cutaneous subset or early disease onset, may identify patients that warrant close monitoring or early and aggressive intervention. In this context, a proper stratification of patients with SSc-ILD is relevant considering most recent data on nintedanib in patients with SSc-associated ILD [52]. Nintedanib effectively slowed progression (decline of forced vital capacity) after initial and long-term use [53], yet without heterogeneity across serological or clinical subgroups [54]. Autoantibodies, such as anti-U11/U12 RNP, anti-Th/To, and others, should be tested on cohorts such as the SENSCIS trial in order to explore their potential to reduce the heterogenicity of SSc-related ILD. This is especially warranted since (a.) the effect of nintedanib was more pronounced in the anti-Scl-70 negative group [54], and (b.) anti-U11/U12 antibodies are more prevalent in CTP negative patients [39]. However, this is not only limited to nintedanib, but applies to other novel and alternative treatment approaches in SSc-ILD, including agents with strong-supporting evidence, such as tocilizumab and rituximab [55,56], or others with biological or promising compoundssuch as abatacept, lenabasum, romilkimab, belimumab, bermekimab, brodalumab, tofacitinib, or even rituximab [reviewed in [1,57]].

## 10. Other Considerations

No data has been published to date on the fluctuation of anti-U11/U12 antibodies over time. Historically, it is established that autoantibody specificities in SSc are rather stable. However, a recent case report on anti-Th/To antibodies has shown significant fluctuations in antibody levels and also seroconversion (switch in autoantibody profile) [58]. The stability of anti-U11/U12 antibody levels might have significant implications on the interpretation of previous data including their association with cancer. Longitudinal studies of large SSc cohorts are required to analyze this aspect of anti-U11/U12 antibodies.

## 11. Conclusions

Although very limited literature is available on anti-U11/U12 autoantibodies, existing evidence strongly indicates that these antibodies are specific for SSc and associated with a severe phenotype with a poor prognosis, mostly characterized by the presence of severe ILD. The reported associations with other disease manifestations, GI involvement, and cancer deserves further studies.

## Figures and Tables

**Figure 2 diagnostics-13-01257-f002:**
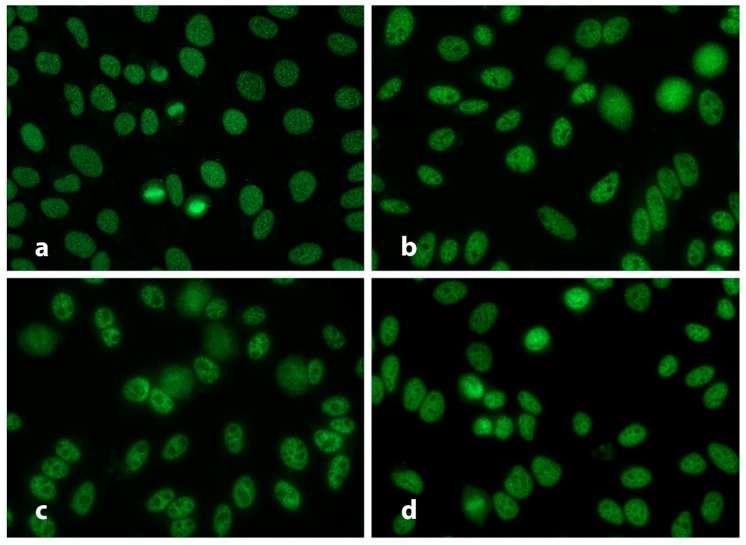
Anti-U11/U12 (RNPC-3) antibodies on HEp-2 substrates by indirect immunofluorescence (IIF) assay. Images of patterns derived from HEp-2 substrates from different manufacturers [(**a**) Bio Rad; (**b**) Inova Diagnostics; (**c**) ImmunoConcepts HEp-2000; (**d**) Binding Site] using an index, monospecific human anti-U11/U12 (RNPC-3) antibody positive serum display fine discrete speckled staining of interphase nuclei. However, depending on the HEp-2 slide manufacturer, variable staining of nucleoli and metaphase chromatin was observed. Original magnification 400×.

**Table 3 diagnostics-13-01257-t003:** Proteins of the U11/U12 complex [32] ordered according to the molecular weight.

Protein ID	Molecular Weight	ID	Alternative Names/Protein Function
hPrp43, DHX15	90 kDa	O43143	ATP-dependent RNA helicase 46, DHX15
RNPC-3	65 kDa	Q96LT9	RNA-Binding Region-Containing Protein 3 RNA-Binding Protein 40 U11/U12 Small Nuclear Ribonucleoprotein 65 kDa Protein
PDCD7	59 kDa	Q8N8D1	Programmed Cell Death Protein 7 U11/U12 SnRNP 59K 2 3
snRNP48	48 kDa	Q6IEG0	
snRNP35	35 kDa	Q16560	
ZCRB1	31 kDa	Q8TBF4	Zinc Finger CCHC-Type And RNA-Binding Motif-Containing Protein 1 U11/U12 Small Nuclear Ribonucleoprotein 31 kDa Protein
snRNP25	25 kDa	Q9BV90	Minus-99 Protein
ZMAT5	20 kDa	Q9UDW3	Zinc Finger Matrin-Type Protein 5 U11/U12 Small Nuclear Ribonucleoprotein 20 kDa Protein
SF3b heptameric complex	N/A	N/A	SF3b1, SF3b2, SF3b3, SF3b4, SF3b5, SF3b6 and SFb7 proteins.
Sm proteins	N/A	N/A	

n/A = not applicable; ID = UniProtKB/Swiss-Prot.

## Data Availability

Not applicable.

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
