# Peer review of "Anti-U11/U12 Antibodies as a Rare but Important Biomarker in Patients with Systemic Sclerosis: A Narrative Review"

_diagnostics, 2023, doi:10.3390/diagnostics13071257_

Round 1

Reviewer 1 Report

The manuscript by Fritzler et al. presents a summary of the current understanding about anti-RNPC-3 antibodies in SSc, including their seroclinical associations, concluding that, although rare and difficult to identify, these antibodies are specific for SSc and associated with a severe phenotype with a poor prognosis mostly characterized by the presence of severe ILD. As a whole, the paper is well structured and there are appropriate and adequate references to related and previous work. Images are explicative and the English language is quite correct. Nevertheless, there are some typos and I have a few comments to rise:  

- Please correct the type of manuscript from Article to Review.

- I think it would be more appropriate to specify in the title that the authors conducted a narrative review. 

- line 56 “wide range of of SSc associated antibodies”: eliminate one of the two “of”.

- the acronyms used throughout the text are not always the same and this makes quite difficult to fully understand the manuscript. Indeed, if the authors initially speak of SSc-SA and SSc-AA referring to systemic sclerosis specific antibodies and to systemic sclerosis associated antibodies, respectively, in the following pages they use different acronyms, i.e. SSA and SAA. I think it would be more appropriate and clearer to always use the same acronym.

- line 69 “consists off several”: please correct with consists of.

- line 70. Please when you first introduce Anti-U11/U12 Ribonucleoprotein (RNP) antibodies at the beginning of the sentence specify that these autoantibodies are also known as anti-RNPC-3.

- line 100 “revealed anti-U11/U12 RNP antibodies were present at low frequency”: please correct with revealed that.

- Paragraph 4.2 Indirect immunofluorescence: please specify what PMAT stands for, as it is the first time this acronym appears.

- line 144 “reports (22, 39, 40)h.”: please eliminate the h.

- line 180 “lablled”: please correct with labelled.

- line 214 “It is clinically important that patients with anti-RNPC-3”. This sentence has no sense.

- line 228 “performed using a 228 cohort (n=447) SSc patients”: please correct with a cohort of.

- line 241 “indiviudals”: please correct with individuals.

- line 278 “In 2016 Xu et used”: please correct with et al.

- line 313 “However, this is not only limit to nintedanib, but expends to other novel and alternative treatment”: please correct with extends.

- Please revise all references throughout the text, as some are in parentheses and not in square brackets (as requested by the journal style).

Author Response

Dear Reviewer, thank you for the careful review of our manuscript. We revised the paper accordingly. 

Reviewer 1

The manuscript by Fritzler et al. presents a summary of the current understanding about anti-RNPC-3 antibodies in SSc, including their seroclinical associations, concluding that, although rare and difficult to identify, these antibodies are specific for SSc and associated with a severe phenotype with a poor prognosis mostly characterized by the presence of severe ILD. As a whole, the paper is well structured and there are appropriate and adequate references to related and previous work. Images are explicative and the English language is quite correct. Nevertheless, there are some typos and I have a few comments to rise:  

- Please correct the type of manuscript from Article to Review.

- I think it would be more appropriate to specify in the title that the authors conducted a narrative review.

Thank you for pointing this out (added to the title).

- line 56 “wide range of of SSc associated antibodies”: eliminate one of the two “of”.

Thank you for pointing this out (corrected).

- the acronyms used throughout the text are not always the same and this makes quite difficult to fully understand the manuscript. Indeed, if the authors initially speak of SSc-SA and SSc-AA referring to systemic sclerosis specific antibodies and to systemic sclerosis associated antibodies, respectively, in the following pages they use different acronyms, i.e. SSA and SAA. I think it would be more appropriate and clearer to always use the same acronym.

- line 69 “consists off several”: please correct with consists of.

Thank you for pointing this out (corrected).

- line 70. Please when you first introduce Anti-U11/U12 Ribonucleoprotein (RNP) antibodies at the beginning of the sentence specify that these autoantibodies are also known as anti-RNPC-3.

Thank you for pointing this out (introduced).

- line 100 “revealed anti-U11/U12 RNP antibodies were present at low frequency”: please correct with revealed that.

Thank you for pointing this out (corrected).

- Paragraph 4.2 Indirect immunofluorescence: please specify what PMAT stands for, as it is the first time this acronym appears.

Thank you for pointing this out (corrected).

- line 144 “reports (22, 39, 40)h.”: please eliminate the h.

- line 180 “lablled”: please correct with labelled

We are sorry, but we can`t find this typo in the text.

- line 214 “It is clinically important that patients with anti-RNPC-3”. This sentence has no sense.

Thank you for pointing this out (sentence has been removed).

- line 228 “performed using a 228 cohort (n=447) SSc patients”: please correct with a cohort of.

Thank you for pointing this out (corrected).

- line 241 “indiviudals”: please correct with individuals.

We are sorry, but we can`t find this typo in the text.

- line 278 “In 2016 Xu et used”: please correct with et al.

Thank you for pointing this out (corrected).

- line 313 “However, this is not only limit to nintedanib, but expends to other novel and alternative treatment”: please correct with extends.

- Please revise all references throughout the text, as some are in parentheses and not in square brackets (as requested by the journal style).

All were now changed to square brackets according to the citation style of the journal

Reviewer 2 Report

Dear Authors, 

The present manuscript compiled the known information about anti-U11/12 antibodies in systemic sclerosis and examined the clinical and serological relationships in detail. Since there are not many studies compiling information on this topic, I consider the current manuscript as a publication that deals with the subject in an original way. In conclusion, it was emphasized that Anti U11/12 antibody is associated with severe ILD and maybe a poor prognostic marker in systemic sclerosis. In addition, I think that the tables and figures used in the study are presented in an explanatory way and the articles that were cited as the references for the information given are appropriate.  

Thanks for your hard work

Author Response

Dear Reviewer, thank you for you for your review

Best wishes